# The AAA ATPase Vps4 binds ESCRT-III substrates through a repeating array of dipeptide-binding pockets

Han Han[†], Nicole Monroe[†], Wesley I Sundquist*, Peter S Shen*, Christopher P Hill*

Department of Biochemistry, University of Utah School of Medicine, Salt Lake City, United States

**Abstract** The hexameric AAA ATPase Vps4 drives membrane fission by remodeling and disassembling ESCRT-III filaments. Building upon our earlier 4.3 Å resolution cryo-EM structure (*Monroe et al., 2017*), we now report a 3.2 Å structure of Vps4 bound to an ESCRT-III peptide substrate. The new structure reveals that the peptide approximates a β-strand conformation whose helical symmetry matches that of the five Vps4 subunits it contacts directly. Adjacent Vps4 subunits make equivalent interactions with successive substrate dipeptides through two distinct classes of side chain binding pockets formed primarily by Vps4 pore loop 1. These pockets accommodate a wide range of residues, while main chain hydrogen bonds may help dictate substrate-binding orientation. The structure supports a 'conveyor belt' model of translocation in which ATP binding allows a Vps4 subunit to join the growing end of the helix and engage the substrate, while hydrolysis and release promotes helix disassembly and substrate release at the lagging end.
DOI: https://doi.org/10.7554/eLife.31324.001

*For correspondence:
wes@biochem.utah.edu (WIS);
peter.shen@biochem.utah.edu
(PSS);
chris@biochem.utah.edu (CPH)

[†]These authors contributed
equally to this work

Competing interest: See
page 11

Reviewing editor: Andreas
Martin, University of California,
Berkeley, United States

## Introduction

The Endosomal Sorting Complexes Required for Transport (ESCRT) pathway drives multiple cellular membrane fission processes (*Christ et al., 2017*; *Scourfield and Martin-Serrano, 2017*) through the formation of filaments comprising different subsets of related ESCRT-III family members. ESCRT-III filaments stabilize highly curved membrane necks that resolve by fission when the filaments are remodeled by Vps4 (*Monroe and Hill, 2016*). Continued Vps4 activity removes ESCRT-III subunits and drives complete filament disassembly, thereby enabling subsequent rounds of ESCRT activity (*Mierzwa et al., 2017*; *Schöneberg et al., 2017*).

Vps4 is monomeric or dimeric at cytoplasmic concentrations, but forms a hexamer when active and recruited to ESCRT-III filaments (*Monroe et al., 2014*). Recruitment is mediated, at least in part, by binding of the Vps4 MIT (Microtubule Interacting and Transport) domains to MIM (MIT Interacting Motif) elements in the exposed tails of ESCRT-III subunits (*Obita et al., 2007*; *Stuchell-Brereton et al., 2007*; *Hurley and Yang, 2008*). Hexamerization is further promoted by the cofactor protein Vta1/LIP5 (*Scott et al., 2005*; *Lottridge et al., 2006*; *Azmi et al., 2008*; *Xiao et al., 2008*), whose VSL domain binds adjacent Vps4 subunits at the ring periphery (*Yang and Hurley, 2010*; *Davies et al., 2014*; *Monroe et al., 2017*; *Sun et al., 2017*). The Vps4 N-terminal MIT domain is followed by an ~40 residue flexible linker and an ATPase cassette that comprises a large ATPase domain, small ATPase domain, and a β domain (*Scott et al., 2005*).

Recently reported cryo-EM structures of Vps4 at overall resolutions of 4.3 Å (*Monroe et al., 2017*), 6.1 Å (*Su et al., 2017*), and 3.9 Å (*Sun et al., 2017*) revealed similar hexameric 'lock washers', in which five of the six Vps4 subunits form a helical assembly and the sixth closes the ring. This arrangement is quite different from the packing seen in multiple Vps4 crystal structures. Although

the three cryo-EM structures are similar, they prompted very different mechanistic models to explain how ESCRT-III subunits are processed. Our structure, which was visualized in complex with a substrate peptide, guided the proposal that ESCRT-III substrates bind within the hexamer pore (*Monroe et al., 2017*), consistent with a model that ESCRT-III subunits are unfolded by translocation through the pore (*Yang et al., 2015*). In contrast, the observation that the Vps4 hexamer adopts 'open' and 'closed' states in the absence of Vta1/LIP5 and substrate, prompted the proposal that substrates are engaged by a single ESCRT-III subunit, and that ATP hydrolysis pulls an entire ESCRT-III subunit from the filament and positions the next Vps4 subunit to remove the ensuing ESCRT-III subunit (*Su et al., 2017*).

We have now determined the Vps4-ESCRT-III$^{peptide}$ complex structure at 3.2 Å resolution. This structure shows that the highly variable side chains of the substrate bind in an equivalent way to an array of two classes of pockets that are repeated throughout the Vps4 pore. Moreover, the ESCRT-III peptide binds in a β-strand conformation in one orientation. These insights support our earlier mechanistic proposal, which may be applicable to other AAA ATPases.

## Results and discussion

### Overall structure

The new reconstruction of the Vps4-Vta1-ESCRT-III$^{peptide}$-ADP·BeF$_x$ complex agrees well with our earlier 4.3 Å resolution reconstruction but now has an overall resolution of 3.2 Å, which likely reflects the use of superior microscope instrumentation (see Materials and methods). The six Vps4 subunits form a closed ring, with six Vta1 VSL domains binding around the periphery and a single ESCRT-III peptide bound in the central pore (*Figure 1*, *Table 1*, *Figure 1—figure supplements 1–4*, *Figure 1—video 1*). Vps4 subunits A-E form a helix whose symmetry approximates a 60° rotation and 6.3 Å translation between adjacent subunits, and provides the binding surface for the ESCRT-III peptide. As noted previously (*Monroe et al., 2017*), subunit E deviates slightly from the more exact helical symmetry of subunits A-D. The large domain of subunit F is disengaged from adjacent Vps4 subunits (and substrate) and appears to be transitioning between the two ends of the Vps4 helix. Subunit F and features at the ring periphery, including the Vps4 β domains and the Vta1 VSL domains, have weak density.

### Nucleotide states and subunit interfaces

Nucleotides primarily contact one Vps4 subunit at a subunit interface, with the β-phosphate and BeF$_x$ contacting two 'finger' arginines, R288 and R289, from the following subunit (*Figure 2*, *Figure 2—videos 1–5*). The bound ADP·BeF$_x$ mimics ATP binding to subunits A, B, and C, whereas subunits D and E appear to bind ADP. The density of subunit F is too weak to reliably assess the presence of nucleotide, and it may be empty. The coordination of bound nucleotides is similar for all Vps4 subunits, and resembles binding to other ATPases (*Wendler et al., 2012*). ADP·BeF$_x$ coordination is essentially identical for A-C and is very similar for the ADP at subunit D, whereas displacement of subunit F results in loss of interaction with the finger arginines for the ADP at subunit E. In contrast to earlier proposals (*Gonciarz et al., 2008*), the hinge angle between large and small domains does not change substantially with the bound nucleotide, being 120–121° for subunits A-D, 117° for subunit E, and 122° for subunit F.

The AB, BC, and CD interfaces are extensive, similar to each other, and include large domain contacts with adjacent large domains and small domains (*Figure 2—videos 6,7*). Comparison of these interfaces with the DE subunit pair shows an ~8° rotation of subunit E which, nevertheless, maintains very similar contacts with the preceding D subunit and relative positions of pore loops (*Figure 2—video 8*). It is therefore uncertain why the AB, BC, and CD interfaces bind ATP (ADP·BeF$_x$) while the similar DE interface appears to bind ADP. Although the density seems clear, we acknowledge that assigning ADP (vs. ADP·BeF$_x$) at this site with absolute certainty will require higher resolution data. Regardless, the finding that subunit E binds ADP at the interface with subunit F further supports our mechanistic model that ATP hydrolysis at subunit D destabilizes the subunit interface to promote formation of the more open nucleotide binding site seen for subunit E (below).

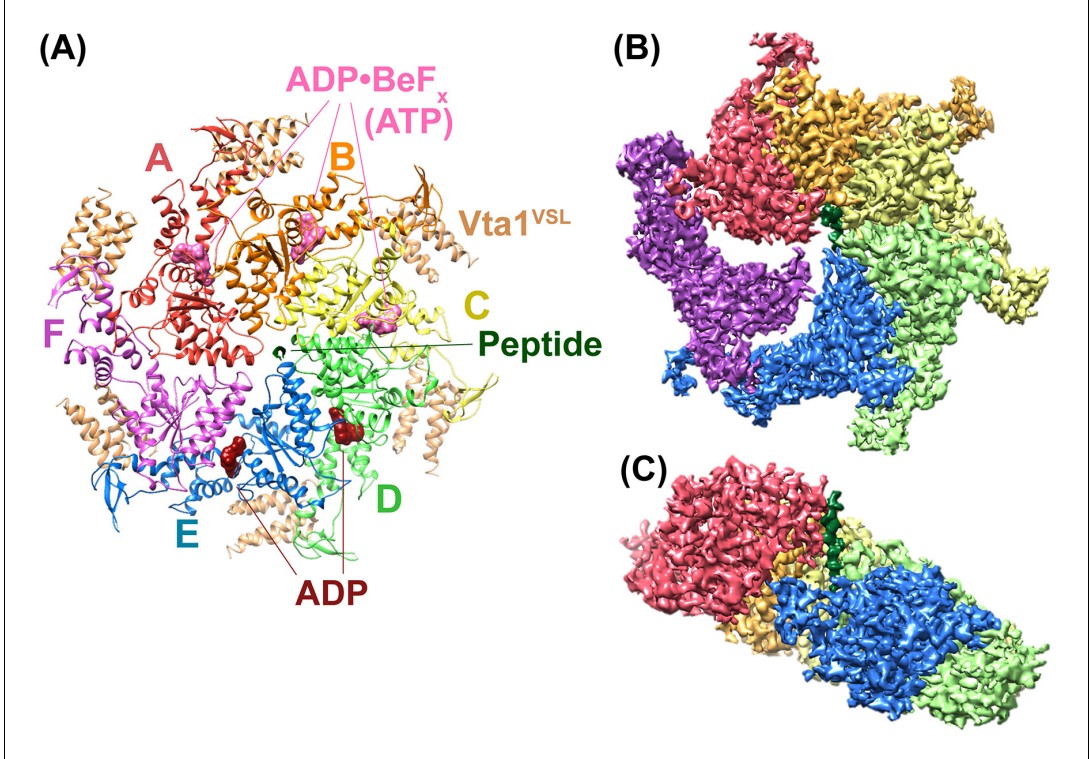

**Figure 1.** Overall structure of the Vps4 complex. (A) Ribbon representation of the complex viewed from the 'top' N-terminal side of Vps4 and N-terminal end of the peptide. (B) Similar orientation as panel A showing a segmented map contoured around Vps4 and peptide. (C) Same as panel B viewed from the side with density for subunit F removed for clarity.

DOI: https://doi.org/10.7554/eLife.31324.002

The following video and figure supplements are available for figure 1:

**Figure supplement 1.** Cryo-EM of the Vps4 complex.

DOI: https://doi.org/10.7554/eLife.31324.003

**Figure supplement 2.** Classification and signal-subtraction scheme for the Vps4 complex.

DOI: https://doi.org/10.7554/eLife.31324.004

**Figure supplement 3.** Focused classification of Vps4 subunit F and Vta1.

DOI: https://doi.org/10.7554/eLife.31324.005

**Figure supplement 4.** Surface Representation.

DOI: https://doi.org/10.7554/eLife.31324.006

**Figure 1—video 1.** Representative density.

DOI: https://doi.org/10.7554/eLife.31324.007

In contrast to the major disruption in contacts between the large domain of subunit F and the large domains of its neighbors, contacts involving the small ATPase domains remain more similar. This is especially true for the FA interface, where the F small domain to A large domain contacts are closely superimposable with those of AB, BC, CD, and DE (*Figure 2—video 9*). The E small domain to F large domain contacts are mediated by the same hydrophobic contacts, albeit with a rotation of ~25° and attendant shifts of 4.5–7.5 Å. Thus, this interaction appears to be maintained throughout the reaction cycle as subunits transition from the lagging (subunit E) to the leading (subunit A) end of the Vps4 helix, and may help maintain the hexameric assembly while the hydrolysis and release of ATP disrupts the core large domain contacts that define the ESCRT-III substrate-binding site.

Interestingly, the small domain-large domain interface is the major lattice contact in all of the reported crystal structures of Vps4, which totals 22 crystallographically unique contacts (*Figure 2—video 10*). The archaeal Vps4 crystal contacts (*Monroe et al., 2014*; *Caillat et al., 2015*) closely resemble the FA/AB/BC/CD/DE interface, while eukaryotic crystal contacts (*Scott et al., 2005*; *Xiao et al., 2007*; *Gonciarz et al., 2008*; *Inoue et al., 2008*) overlap with the EF interface, except

**Table 1.** Reconstruction, Refinement, and Model Statistics of Vps4.

| Reconstruction | |
|---|---|
| Particle images | 82,225 |
| Resolution (0.143 FSC) (Å) | 3.2 |
| Map sharpening B-factor (Å$^2$) | −125 |
| EMDB accession number | EMD-8887 |
| Model refinement and validation of Vps4 subunits A-E | |
| PDB accession number | 6BMF |
| Resolution used for refinement (Å) | 3.2 |
| Number of atoms | 11033 |
| RMSD | |
| Bond length (Å) | 0.01 |
| Bond angles (°) | 0.18 |
| Ramachandran | |
| Favored (%) | 89.5 |
| Allowed (%) | 10.5 |
| Outlier (%) | 0 |
| Validation scores | |
| Molprobity score/percentile (%) | 1.83 (100%) |
| Clashscore/percentile (%) | 5.02 (100%) |
| EMRinger score | 2.04 |

DOI: https://doi.org/10.7554/eLife.31324.008

two of the three mouse Vps4 crystal contacts (*Inoue et al., 2008*), which are intermediate between the two states. These differences correlate with the presence of a Vps4 β domain and Vta1 cofactor in eukaryotes but not archaea, supporting the interpretation that Vta1 promotes formation and maintenance of the closed helical substrate-binding assembly of Vps4.

## Coordination of the ESCRT-III peptide

The DEIVNKVL ESCRT-III peptide is clearly defined in the new 3.2 Å map (*Figure 3—video 1*), with the exception of the first two side chains, D1 and E2 (Vps2 D165 and E166), which have weak density, as is typically seen for carboxylates in cryo-EM maps (*Mitsuoka et al., 1999*; *Bartesaghi et al., 2014*; *Yonekura et al., 2015*; *Hryc et al., 2017*). The assigned peptide orientation was validated by building and refining in the reverse direction, which showed a correlation coefficient of 0.85 for the assigned orientation vs. 0.81 in the reversed orientation, and EMRinger (*Barad et al., 2015*) scores of 3.7 for the assigned orientation vs. 1.1 for the reversed orientation (*Figure 3—figure supplement 1*).

The peptide adopts an extended conformation that resembles a canonical β-strand (phi −92° to −151°; psi 102° to 189° (−171°)) and packs closely against Vps4 subunits A-E (*Figure 3*, *Figure 3— videos 2–4*). Two distinct classes of side chain binding sites propagate along the pore. Odd-numbered ESCRT-III residues (D1, I3, N5, V7) bind in 'class I' pockets, while the side chains of even-numbered ESCRT-III residues (E2, V4, K6, L8) bind in 'class II' pockets. Class I pockets are formed by pore loop 1 residues K205 and W206, with substrate side chains sandwiched between W206 from successive subunits. The pocket is flanked by K205 from the first Vps4 subunit, which may also stabilize the ladder of W206 side chains through cation-π interactions. .

The class II pockets are formed by the pore loop 1 M207 residues of successive subunits, and are flanked by pore loop 2 residues of the first Vps4 subunit and, to a greater extent, by the pore loop 2 residues of the preceding subunit. The first of these pockets, which includes M207 residues of the A and B subunits, is incompletely formed because the pore loop 2 residues of the preceding subunit (F) are not yet in position. The three pore loop 2 residues that most closely approach the peptide (E245, S246, E247) do not make distinctive contacts with the ESCRT-III peptide. S246 caps the

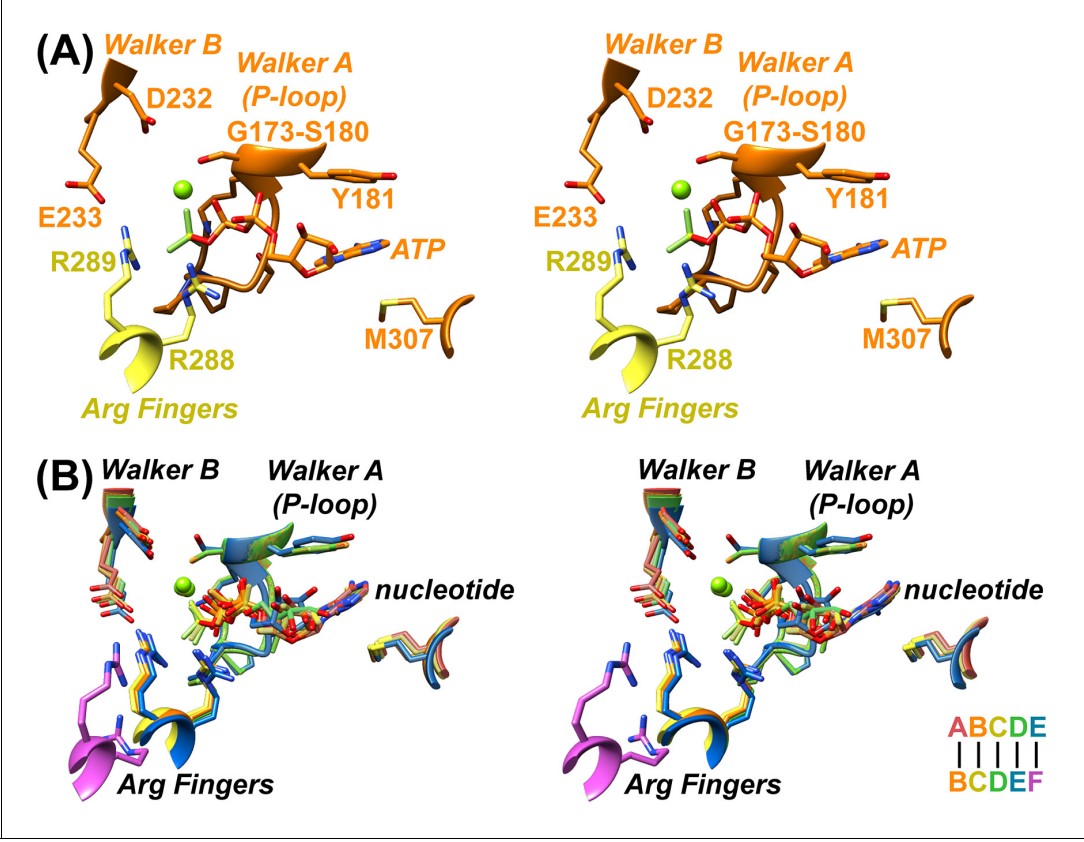

**Figure 2.** Nucleotide coordination and subunit interfaces. (**A**) Stereoview of a representative ADP·BeF$_x$ coordination shown at subunit B (BC interface). Subunits color-coded as in *Figure 1*. (**B**) Stereoview of nucleotide-binding sites at subunits A, B, C, D, and E following superposition on the large domains of the first subunit at each interface.

DOI: https://doi.org/10.7554/eLife.31324.009

The following videos are available for figure 2:

**Figure 2—video 1.** Nucleotide densities.
DOI: https://doi.org/10.7554/eLife.31324.010

**Figure 2—video 2.** Coordination of ADP·BeF$_x$ (ATP) at a representative subunit.
DOI: https://doi.org/10.7554/eLife.31324.011

**Figure 2—video 3.** Comparison of nucleotide coordination at subunits A, B, and C.
DOI: https://doi.org/10.7554/eLife.31324.012

**Figure 2—video 4.** Comparison of nucleotide coordination at subunits A and D.
DOI: https://doi.org/10.7554/eLife.31324.013

**Figure 2—video 5.** Comparison of nucleotide coordination at subunits A and E.
DOI: https://doi.org/10.7554/eLife.31324.014

**Figure 2—video 6.** Interface at a representative subunit pair in the Vps4 helix.
DOI: https://doi.org/10.7554/eLife.31324.015

**Figure 2—video 7.** Comparison of AB, BC, and CD interfaces.
DOI: https://doi.org/10.7554/eLife.31324.016

**Figure 2—video 8.** Comparison of AB and DE interfaces.
DOI: https://doi.org/10.7554/eLife.31324.017

**Figure 2—video 9.** Conservation of the interface between small and large domains.
DOI: https://doi.org/10.7554/eLife.31324.018

**Figure 2—video 10.** Similarity between the small domain - large domain interfaces and the major contacts in Vps4 crystal structures.
DOI: https://doi.org/10.7554/eLife.31324.019

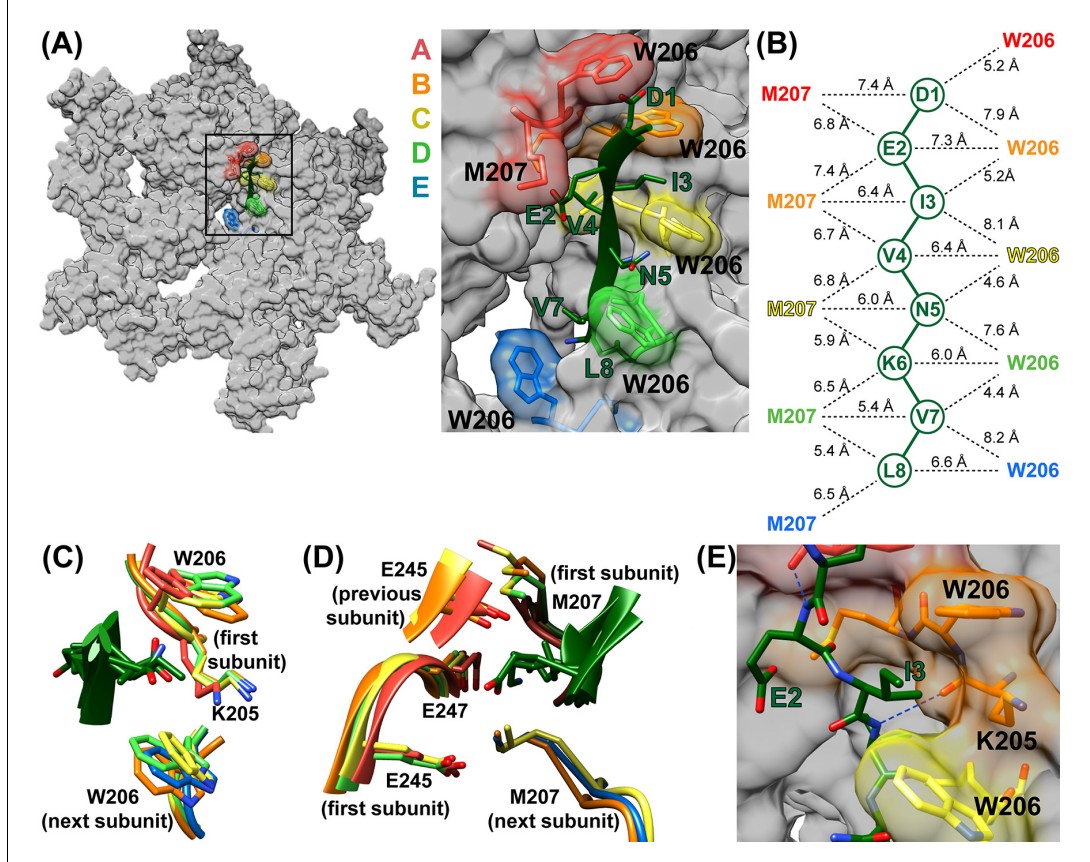

**Figure 3.** ESCRT-III peptide conformation and coordination. (**A**) Left – tilted view of a surface representation showing how the pore loop residues form an array of class I and II binding pockets through the hexamer pore. W206 and M207 from subunits A-E are highlighted. Right – close up of the pore region. (**B**) Distances between Cα atoms of the peptide and pore loop 1 W206 and M207 indicate equivalent binding in the different class I and class II pockets. (**C**) Superposition of the four Class I pockets following superposition on Cα atoms of the class I pocket residues of subunits A and B. (**D**) Superposition of the four class II pockets following superposition on Cα atoms of the class II pocket residues of subunits A and B. (**E**) The H-bond seen between the NH of even-numbered ESCRT-III residues and the K205 CO of Vps4 subunits A-D – here centered on the bond between ESCRT-III V4 and subunit B. The bond between E2 and subunit A is also visible.

DOI: https://doi.org/10.7554/eLife.31324.020

The following video and figure supplement are available for figure 3:

**Figure supplement 1.** Fit of peptide to density when refined in the assigned and reversed orientations.
DOI: https://doi.org/10.7554/eLife.31324.021
**Figure 3—video 1.** Charge density map at the ESCRT-III peptide and residues of pore loop 1 and 2.
DOI: https://doi.org/10.7554/eLife.31324.022
**Figure 3—video 2.** The ESCRT-III peptide spirals around the helix axis.
DOI: https://doi.org/10.7554/eLife.31324.023
**Figure 3—video 3.** Class I side chain binding pockets.
DOI: https://doi.org/10.7554/eLife.31324.024
**Figure 3—video 4.** Class II side chain binding pockets.
DOI: https://doi.org/10.7554/eLife.31324.025

N-terminus of a Vps4 helix while E245 and E247 make non-specific contacts with peptide side chains. The different class I and class II pockets are all essentially identical (*Figure 3CD*) and their exposure to the highly solvated pore explains how polar side chains can be accommodated.

The refined model indicates that the NH groups of the even-numbered ESCRT-III peptide residues form hydrogen bonds with the main chain O of Vps4 K205 (subunit A, B, C, and D distances are 3.1–3.5 Å). In contrast, these distances are 3.4–4.5 Å in the model refined with the peptide in

the reverse orientation. Thus, substrate NH hydrogen bonds may help define the substrate orientation by optimally positioning ESCRT-III side chains with respect to their binding pockets (*Figure 3E*).

## Model of translocation

The structure supports our earlier conveyor-belt model of substrate translocation (*Monroe et al., 2017*) but now includes more detail, especially of substrate binding. It is also consistent with other recent models (*Gates et al., 2017*; *Monroe et al., 2017*; *Puchades et al., 2017*; *Ripstein et al., 2017*), albeit with additional molecular details. We envision that the enzyme proceeds by transitioning of subunit configurations around the hexameric ring, such that one step represents transitioning of subunits F, A, B, C, D, E to the configurations of subunits A, B, C, D, E, F (*Figure 4A*, *Figure 4— video 1*). Each step comprises concomitant changes in subunit interfaces at each end of the Vps4 A-E helix. At the leading end, binding of ATP allows subunit F to pack against subunit A and thereby bind the next two residues of the substrate. At the lagging end, hydrolysis of ATP and subsequent phosphate release destabilizes the interface and drives subunit E to the transitioning configuration, thereby opening the nucleotide binding site to allow exchange of ADP by ATP. In this manner, the equivalent dipeptide-binding sites formed at the interfaces of subunits A-E track with their bound dipeptides along the pore as subunits transition from the lagging end to the leading end of the helix. Thus, the pathway can be represented either as Vps4 'walking' along the substrate or as substrate being translocated through the Vps4 pore, depending upon the frame of reference.

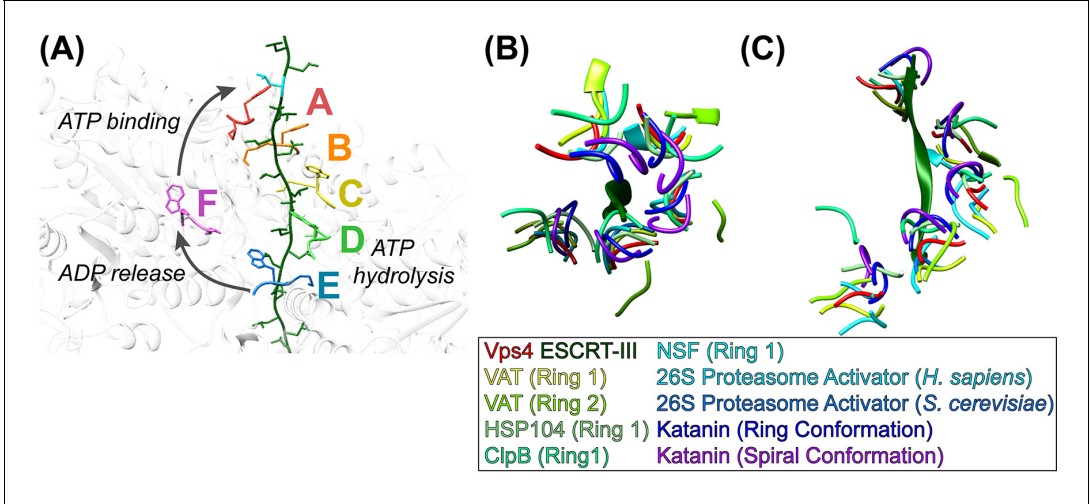

**Figure 4.** Mechanism of translocation. (**A**) Proposed mechanism of ESCRT-III translocation by Vps4. W206 and M207 residues of the six Vps4 subunits are shown, with the peptide passing through the Vps4 hexamer. The peptide model was constructed by changing the side chains to leucine without adjusting the main chain, and building out in the N and C directions by overlapping copies of the peptide model. The proposed mechanism envisions that Vps4 progresses through states A to E while bound to successive dipeptides of its substrate. ATP hydrolysis at subunit D destabilizes the DE interface and promotes displacement of subunit E toward the transitioning subunit F configuration, which allows displacement of ADP. Subsequent ATP binding allows subunit F to pack against subunit A, bind to the next dipeptide of ESCRT-III, and assume the subunit A configuration. (**BC**) Conservation of helical pore loop structure in AAA ATPases. Overlap on the large ATPases of multiple AAA ATPase structures gives a similar helical arrangement of pore loop 1 residues from five subunits. (**B**) Top and (**C**) side views are shown of the ESCRT-III peptide (green) and Vps4 pore loop 1 (red) with the equivalent residues of: VAT (*Ripstein et al., 2017*) (pdbid 5vca), HSP104 (*Gates et al., 2017*) (5vjh), NSF (*Zhao et al., 2015*) (3j94), human 26S proteasome (*Huang et al., 2016*) (5gjr), yeast 26S proteasome (*Wehmer et al., 2017*) (5mp9), katanin (*Zehr et al., 2017*) (5wc0, 5wcb).
DOI: https://doi.org/10.7554/eLife.31324.026

The following video and figure supplement are available for figure 4:

**Figure supplement 1.** Structure-based alignment of pore loop 1 sequences.
DOI: https://doi.org/10.7554/eLife.31324.027
**Figure 4—video 1.** Model of translocation.
DOI: https://doi.org/10.7554/eLife.31324.028
**Figure 4—video 2.** Comparison with other classic clade AAA+ ATPases.
DOI: https://doi.org/10.7554/eLife.31324.029

Although our structure shows four intact dipeptide-binding sites and just one transitioning subunit, it is expected that two Vps4 subunits will be disengaged from substrate through at least part of the reaction cycle. This is because subunit E will disengage from substrate as it moves toward the F configuration while the F subunit is moving toward the substrate-binding A configuration. The specific point in the reaction cycle captured in our structure corresponds to subunit E just starting to transition toward the F state, as indicated by the slight deviation of the DE interface from the configuration of AB, BC, and CD, and consistent with our structural interpretation that ADP is bound to subunit D (above). The stability of this state relative to other conformations along the reaction coordinate will result from a combination of multiple factors, including the particular nucleotide (ADP·BeF$_x$) and ESCRT-III peptide in the complex.

Our model implies that each step translocates two amino acid residues and hydrolyses one ATP molecule. It also explains the importance of the pore loop residues and the integral role of the class I and II pockets, because once a substrate residue binds at the top of the Vps4 helix it does not substantially change conformation until released at the bottom of the helical conveyor belt. The model is also consistent with reports that other AAA+ ATPases bind a maximum of three or four ATP molecules (*Hersch et al., 2005*; *Horwitz et al., 2007*; *Yakamavich et al., 2008*; *Smith et al., 2011*). Moreover, despite important structural differences, our mechanistic model is analogous to mechanisms established for translocation of DNA by the E1 helicase (*Enemark and Joshua-Tor, 2006*) and of RNA by the Rho translocase (*Thomsen and Berger, 2009*; *Thomsen et al., 2016*).

## Comparison with other AAA ATPases

Vps4 is a member of the classic clade of AAA+ ATPases, which includes the original members of the AAA ATPase family (*Iyer et al., 2004*; *Erzberger and Berger, 2006*). These proteins are hexameric protein translocases, whose conserved pore loops emanate from equivalent structural elements, although their N-terminal domains are variable and only Vps4 has a β domain. Family members are found in a variety of contexts. For example, Vps4 is a homohexamer that forms a single ring, whereas p97/CDC48/VAT and NSF comprise two AAA ATPase cassettes that each form a hexameric ring. Others, such as HSP104/ClpB, form a double ATPase ring structure in which only one of the rings belongs to the classic clade. Still others, such as the eukaryotic proteasome ATPases, form part of a much larger complex and comprise six different, albeit related, subunits.

Sequence alignment of classic clade AAA+ ATPase family members for which structures have been reported shows that the first two of the three pore loop 1 residues that contact substrate (Vps4 K205, W206, M207) are conserved (*Figure 4—figure supplement 1*). W206 is always W, F, or Y, all of which could perform the role of sandwiching substrate side chains. The preceding K205 residue is usually K but sometimes M, which like K could flank the class I binding pocket. The more variable third residue, M207, is usually I, L or V, but there are examples of M (Vps4), A, F, K, R, and Y, all of which could provide hydrophobic sides to the class II binding pocket. Interestingly, the following residue, G208, is invariant, presumably because it adopts phi angles (~81 to 85°) that are only favored for glycine, and helps define a conformation that can pack against pore loop 1 residues from neighboring subunits. Pore loop 2 residues are less conserved and are typically disordered in other published structures, which is consistent with the lack of distinctive roles in contacting substrate.

Our Vps4 structure superimposes closely with the recently reported structures of VAT (*Ripstein et al., 2017*), HSP104 (*Gates et al., 2017*), and ClpB (*Deville et al., 2017*), which were each determined with substrate bound in their central pores, albeit at relatively low resolutions. A very recently published 3.4 Å structure of the YME1 AAA ATPase in complex with a mixed polypeptide substrate (*Puchades et al., 2017*) also presents a very similar structure and mechanism to that described here, although coordinates are not yet available during final preparation of this manuscript. Close superposition is also seen with multiple other classic clade AAA+ ATPases that have been visualized in the absence of substrate (*Figure 4B*, *Figure 4—video 2*). Notable exceptions include p97 and CDC48, whose structures display rotational rather than helical symmetry (*Banerjee et al., 2016*; *Xia et al., 2016*), and ClpX and HslU, which are not members of the classic clade but share notable similarities with Vps4 and have provided leading mechanistic models for protein translocating AAA+ ATPases (*Olivares et al., 2016*).

In contrast to these sequential models, biochemical analyses of ClpX (*Martin et al., 2005*) and HslU (*Baytshtok et al., 2017*) have argued for stochastic mechanisms by showing that just one active

ATPase subunit per hexamer can drive translocation, albeit at much reduced efficiency. A possible resolution is that a hexamer that has only one active ATPase site may allow inactive subunits to diffuse through the entire helical cycle. Thus, like a 6-cylinder engine firing on just one cylinder, a single active ATPase might drive the sequential conveyor-belt model, albeit rather poorly.

In summary, the structure shows details of substrate interactions that are provided by a repeating array of dipeptide binding sites with the ESCRT-III peptide in a unique orientation. It further supports a sequential mechanism and explains the ability to translocate polypeptides with little sequence specificity. Important future priorities include testing the generality of the structural observations, mechanistic implications, and the extent to which they may apply to other AAA+ ATPases.

## Materials and methods

### Electron microscopy

Sample preparation was as described (*Monroe et al., 2017*). Vitrified grids were loaded onto a Titan Krios (FEI) operating at 300 kV. Images were acquired using a defocus range between −1.0 to −2.2 μm. A total of 2,349 cryo-EM movies were recorded using a K2 Summit direct detector (Gatan) in counting mode with a pixel size of 1.10 Å and at a dose rate of ~7.4 $e^-$/pixel/sec. Each movie was recorded as a stack of 40 frames accumulated over 10 s, totaling ~62$e^-$/Å$^2$.

### Cryo-EM analysis

Movie frames were aligned, dose weighted, and summed using MotionCor2 (*Zheng et al., 2017*) (*Figure 1—figure supplement 1A*). CTF parameters were determined on non-dose-weighted sums using gctf (*Zhang, 2016*). Micrographs with poor CTF cross correlation scores were excluded from downstream analyses. A total of 1,987 dose-weighted sums were used for all subsequent image processing steps. 4,429 particles were manually selected from 30 micrographs in EMAN2 using the e2boxer.py program (*Tang et al., 2007*) to generate preliminary 2D classes in RELION (*Scheres, 2012*). The non-CTF-corrected class averages were used for template-based autopicking in gautomatch. A total of 599,085 particles were extracted and used as input for full CTF-corrected image processing (*Figure 1—figure supplement 1B*). After multiple rounds of 2D classification, 124,743 particles were retained based on visual inspection of classes with high-resolution Vps4 features and used for an initial round of 3D classification. After 3D classification, 109,241 particles were used for RELION auto-refinement (*Scheres, 2012*), which generated a 4.1 Å density map of the Hcp1-Vps4 fusion complex based on the gold-standard FSC criterion (*Figure 1—figure supplement 1D*). To improve the resolution of Vps4, we performed signal subtraction of Hcp1 densities using the same approach as described previously (*Bai et al., 2015*; *Monroe et al., 2017*) (*Figure 1—figure supplement 2*). After Hcp1 signal subtraction, we performed an additional round of 3D classification, which assigned 82,225 particles into a single class with excellent Vps4 features. These particles were used for a final round of RELION auto-refinement, producing a 3.2 Å resolution density map of Vps4 (*Figure 1—figure supplement 2*). B-factor sharpening of −125 Å$^2$ was applied using an automated procedure in RELION postprocessing (*Rosenthal and Henderson, 2003*). Local resolutions were estimated using ResMap (*Kucukelbir et al., 2014*) (*Figure 1—figure supplement 1G*).

The consensus reconstruction of the Vps4 complex revealed poor, fragmented densities for subunit F and Vta1. To improve their densities, we performed additional rounds of focused classification by generating custom soft-edged masks around their respective densities and then using RELION to classify the particles without re-alignment (*Figure 1—figure supplement 3*). Particles from classes with ordered densities were used for separate RELION auto-refinement reconstructions and produced lower-resolution maps that were used for rigid-body fitting of subunit F or Vta1.

### Model building, refinement and validation

Model building and refinement followed the same approach as for the earlier lower resolution structure (*Monroe et al., 2017*). NCS restraints were applied to Vps4 subunits A-E with the exception of residues 204–208, 227–233, and 249–271 of subunit A and residues 140–158, 171–191, and 204–205 of subunit E. For subunits A, B, C, and D, the distance between the Mg$^{2+}$ and the OG of 180S was restrained to 2.0 Å. No reference model was used during refinement. The refined model was assessed using MolProbity (RRID: SCR_014226) (*Chen et al., 2010*) and EMRinger (*Barad et al.,*

*2015*). To test for overfitting, all atoms in the refined model were randomly displaced by 0.5 Å and re-refined against one of the RELION half maps. FSC curves were generated for the re-refined model against the half map used for re-refinement (FSC$_{work}$) and against the other half map (FSC$_{test}$). The close agreement between the two curves is consistent with lack of overfitting (*Figure 1—figure supplement 1E*).

To validate the orientation of the 8-residue ESCRT-III peptide, it was built and refined in opposing conformations (*Figure 3—figure supplement 1*). RSCC scores between the models and density map were determined using UCSF Chimera (*Pettersen et al., 2004*). Side chain-directed model versus map calculations for the peptide were performed using EMRinger (*Barad et al., 2015*).

### Structure deposition

The refined model comprising the Vps4 ATPase domains of subunits A-E and ESCRT-III peptide has been deposited into the PDB (RRID: SCR_012820; PDB ID: 6BMF). The complete model, including regions not subjected to atomic refinement such as the 12 Vta1$^{VSL}$ domains and subunit F, has been deposited into the PDB (PDB ID: 6AP1) together with the sharpened Hcp1-subtracted map (RRID: SCR_003207, EMDB Accession Number EMD-8887). The unsharpened map, the two maps for subunit F, and the six maps for the Vta1 VSL domain have been deposited at the EMDB (RRID: SCR_003207, EMDB Accession Number(s) EMD-8888, EMD-8889, EMD-8890, EMD-8891, EMD-8892, EMD-8893, EMD-8894, EMD-8895, EMD-8896).

## Acknowledgements

Electron microscopy was performed at the National Resource for Automated Molecular Microscopy and the Simons Electron Microscopy Center located at the New York Structural Biology Center, supported by grants from NIH (GM103310, S10 OD019994) and the Simons Foundation (349247), with additional support from the Agouron Institute (F00316). We thank Zhening Zhang for assistance in data collection. The Center for High Performance Computing at the University of Utah provided computing resources. We thank James Fulcher and Michael Kay for peptide synthesis, and Frank Whitby for help with model refinement.

## Additional information

### Competing interests

Wesley I Sundquist: Reviewing editor, *eLife*. The other authors declare that no competing interests exist.

### Funding

| Funder | Grant reference number | Author |
|---|---|---|
| National Institutes of Health | P50 GM082545 | Han Han<br>Nicole Monroe<br>Wesley I Sundquist<br>Peter S Shen<br>Christopher P Hill |
| National Institutes of Health | T32 AI055434 | Nicole Monroe |
| National Institutes of Health | R37 AI051174-16 | Nicole Monroe<br>Wesley I Sundquist |
| National Institutes of Health | R01 GM112080 | Nicole Monroe<br>Wesley I Sundquist |

The funders had no role in study design, data collection and interpretation, or the decision to submit the work for publication.

### Author contributions

Han Han, Formal analysis, Validation, Investigation, Visualization; Nicole Monroe, Formal analysis, Investigation, Visualization, Writing—review and editing; Wesley I Sundquist, Conceptualization,

Resources, Supervision, Funding acquisition, Project administration, Writing—review and editing; Peter S Shen, Data curation, Formal analysis, Validation, Investigation, Visualization, Writing—review and editing; Christopher P Hill, Conceptualization, Resources, Supervision, Funding acquisition, Validation, Writing—original draft, Project administration

**Author ORCIDs**
Han Han (iD) http://orcid.org/0000-0003-0361-4254
Nicole Monroe (iD) http://orcid.org/0000-0001-7678-4997
Wesley I Sundquist (iD) http://orcid.org/0000-0001-9988-6021
Peter S Shen (iD) http://orcid.org/0000-0002-6256-6910
Christopher P Hill (iD) http://orcid.org/0000-0001-6796-7740

**Decision letter and Author response**
Decision letter https://doi.org/10.7554/eLife.31324.058
Author response https://doi.org/10.7554/eLife.31324.059

# Additional files

## Supplementary files
• Supplementary file 1. MolProbity report. This is for the parts of the model that were defined in charge density at a resolution that justified refinement (Subunits A-E, nucleotides, ESCRT-III peptide). Data in *Table 1* are based on this report.
DOI: https://doi.org/10.7554/eLife.31324.030
• Transparent reporting form
DOI: https://doi.org/10.7554/eLife.31324.031

## Major datasets
The following datasets were generated:

| Author(s) | Year | Dataset title | Dataset URL | Database, license, and accessibility information |
|---|---|---|---|---|
| Han H, Monroe N, Shen P, Sundquist WI, Hill CP | 2017 | Vps4p-Vta1p complex with peptide binding to the central pore of Vps4p | http://www.rcsb.org/pdb/explore/explore.do?structureId=6AP1 | Publicly available at the RCSB Protein Data Bank (accession no. 6AP1) |
| Han H, Monroe N, Shen P, Sundquist WI, Hill CP | 2017 | Vps4p-Vta1p complex with peptide binding to the central pore of Vps4p | http://www.ebi.ac.uk/pdbe/entry/emdb/EMD-8887 | Publicly available at the EMDataBank (accession no. EMD-8887) |
| Han H, Monroe N, Shen P, Sundquist WI, Hill CP | 2017 | Unsharpened map of Vps4p-Vta1p complex with peptide binding to the central pore of Vps4p | http://www.ebi.ac.uk/pdbe/entry/emdb/EMD-8888 | Publicly available at the EMDataBank (accession no. EMD-8888) |
| Han H, Monroe N, Shen P, Sundquist WI, Hill CP | 2017 | Focused classification map for high position subunit F of Vps4p-Vta1p complex with peptide binding to the central pore of Vps4p | http://www.ebi.ac.uk/pdbe/entry/emdb/EMD-8889 | Publicly available at the EMDataBank (accession no. EMD-8889) |
| Han H, Monroe N, Shen P, Sundquist WI, Hill CP | 2017 | Focused classification map for low position subunit F of Vps4p-Vta1p complex with peptide binding to the central pore of Vps4p | http://www.ebi.ac.uk/pdbe/entry/emdb/EMD-8890 | Publicly available at the EMDataBank (accession no. EMD-8890) |
| Han H, Monroe N, Shen P, Sundquist WI, Hill CP | 2017 | Focused classification map for VSL dimer bridging Subunit A and B of Vps4p-Vta1p complex with peptide binding to the central pore of Vps4p | http://www.ebi.ac.uk/pdbe/entry/emdb/EMD-8891 | Publicly available at the EMDataBank (accession no. EMD-8891) |

| Han H, Monroe N, Shen P, Sundquist WI, Hill CP | 2017 | Focused classification map for VSL dimer bridging Subunit B and C of Vps4p-Vta1p complex with peptide binding to the central pore of Vps4p | http://www.ebi.ac.uk/pdbe/entry/emdb/EMD-8892 | Publicly available at the EMDataBank (accession no. EMD-8892) |
|---|---|---|---|---|
| Han H, Monroe N, Shen P, Sundquist WI, Hill CP | 2017 | Focused classification map for VSL dimer bridging Subunit C and D of Vps4p-Vta1p complex with peptide binding to the central pore of Vps4p | http://www.ebi.ac.uk/pdbe/entry/emdb/EMD-8893 | Publicly available at the EMDataBank (accession no. EMD-8893) |
| Han H, Monroe N, Shen P, Sundquist WI, Hill CP | 2017 | Focused classification map for VSL dimer bridging Subunit D and E of Vps4p-Vta1p complex with peptide binding to the central pore of Vps4p | http://www.ebi.ac.uk/pdbe/entry/emdb/EMD-8894 | Publicly available at the EMDataBank (accession no. EMD-8894) |
| Han H, Monroe N, Shen P, Sundquist WI, Hill CP | 2017 | Focused classification map for VSL dimer bridging Subunit E and F of Vps4p-Vta1p complex with peptide binding to the central pore of Vps4p | http://www.ebi.ac.uk/pdbe/entry/emdb/EMD-8895 | Publicly available at the EMDataBank (accession no. EMD-8895) |
| Han H, Monroe N, Shen P, Sundquist WI, Hill CP | 2017 | Focused classification map for VSL dimer bridging Subunit F and A of Vps4p-Vta1p complex with peptide binding to the central pore of Vps4p | http://www.ebi.ac.uk/pdbe/entry/emdb/EMD-8896 | Publicly available at the EMDataBank (accession no. EMD-8896) |
| Han H, Monroe N, Shen P, Sundquist WI, Hill CP | 2017 | Vps4p-Vta1p complex with peptide binding to the central pore of Vps4p | http://www.rcsb.org/pdb/explore/explore.do?structureId=6BMF | Publicly available at the RCSB Protein Data Bank (accession no. 6BMF) |

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
