## [Decision Letter]

Thank you for submitting your article "The AAA ATPase Vps4 binds substrates through a repeating array of dipeptide-binding pockets" for consideration by *eLife*. Your article has been favorably evaluated by Michael Marletta (Senior Editor) and three reviewers, one of whom, Andreas Martin (Reviewer #1), is a member of our Board of Reviewing Editors. The following individual involved in review of your submission has agreed to reveal their identity: Gabriel C Lander (Reviewer #3).

The reviewers have discussed the reviews with one another and the Reviewing Editor has drafted this decision to help you prepare a revised submission.

Summary:

In this manuscript, Sundquist, Hill, and colleagues present the 3.2 Å cryo-EM structure of yeast Vps4 bound to an ESCRT-III peptide. It builds on their paper published in *eLife* earlier this year that provided a 4.3 Å structure of Vps4 bound to nucleotides and a substrate peptide, and proposed a spiral staircase model for the continuous conveyance of substrate through the central pore. In this previous structure, the peptide density showed contacts with multiple subunits via their pore loops, but did not reveal side-chain density or the orientation of the bound peptide.

A Krios data set now improved the resolution to 3.2 Å, which allowed the visualization of peptide side chains as well as pore loop contacts, and the proposal of a more detailed translocation model where every Vps4 subunit interacts with two amino acids through two "binding pockets" formed by the pore-1 and pore-2 loops. The new structure is thus consistent with the mechanisms the authors proposed previously, but does not reveal major new conceptual insights.

The reviewers were a bit surprised to see that there are really not many new data in this submission albeit now having a structure at 1.1 Å higher resolution, and that the authors did not take advantage of this higher resolution to address and answer important outstanding questions.

Even though it is exciting to directly visualize substrate interactions of Vps4, the advance by this manuscript in its current form is rather limited and considered not sufficient for publication in *eLife*. The higher resolution data should enable the authors to pursue further functional exploration and reveal new conceptual insights that go beyond of what had already been established previously for Vps4 and other AAA motors. The manuscript thus needs major revisions before publication.

Essential revisions:

1) The proposal of a directionality for ESCRT substrate translocation by Vps4 is interesting, but should be further supported by experimental data. Several related AAA motors have been shown to not have a preference for substrate orientation in the central pore, and for the conclusions here it should be considered that the described trapped structure likely represents an energy minimum, where the Vps4 subunits might arrange around the stably bound peptide. It is possible that a peptide bound in the opposite orientation could induce equivalent contacts, and that there is in fact no preference in the direction of translocation. The authors may be able to explore this through biochemical assays or structurally, for instance by using peptides of different lengths or with the inverted amino-acid sequence relative to the current one.

2) The presented translocation model also does not go significantly beyond what had been previously discussed. The authors describe a correlation between nucleotide state and subunit conformation relative to the substrate, but the structural relationship remains unexplored in the manuscript. The impressive resolution of this structure should enable the authors to explain how ATP hydrolysis and release translates to specific rearrangements of the subunits or pore loops. Reviewers were also not fully convinced by the authors' assignment of ADP- vs. ATP- (or ADP·BeF_x_) bound subunits, especially for the DE interface, and the presented model for ATPase-driven substrate translocation. As the authors know, reliably distinguishing between nucleotide states has been very difficult for other AAA hexamers in recent high-resolution cryo-EM studies, in part due to weaker density for beta and gamma phosphates, but also due to issues when averaging thousands of particles with potentially distinct nucleotide occupancies in some of the ATPase subunits. For the present study, it is indeed quite unexpected that neither the Mg ion nor the P-loop residues show significant changes in their orientation, and that the Arg fingers are just slightly shifted when comparing supposedly ADP·BeF_x_ -bound and ADP-bound subunits. When discussing this surprising similarity of subunit interfaces, the authors bring up the possibility that their nucleotide assignments are wrong. The authors also try to come up with a rationale for how the very small structural differences may lead to preferential binding of ADP vs. ATP (or ADP·BeF_x_), but ignore in their Discussion that during normal motor function the nucleotide dictates subunit conformation, not the other way around. Several biochemical and structural studies previously suggested major conformational and functional differences between ATP- and ADP-subunits, and that the transition from ATP- to ADP-bound through hydrolysis and phosphate release may drive the power stroke of AAA motors. Interestingly, the authors claim that substantial conformational adjustments are not required to pass through the DE states. But what then is the force-generating step for translocation, if subunits can transition from ATP to ADP-bound without any major conformational changes? ADP release or ATP binding to an empty Vps4 subunit? In a revised manuscript, the authors should assess and discuss which step of the ATPase cycle likely induces conformational changes and drives the conveyer belt for forceful substrate translocation. In that context it should also be discussed how constrains in the topologically closed ring could restrict conformational changes in response to hydrolysis, which may explain the authors' findings that ADP and ADP·BeF_x_-bound subunits don't show major conformational differences.

3) The reviewers have expressed concerns regarding the atomic model refinement. Only statistics for chains A-E are listed in Table 1, but the deposited models contain all 6 subunits in addition to the substrate. The validation report from the PDB for the full deposition reveals statistics that are much worse than those reported in Table 1, and the selective inclusion of only chain A-E is rather misleading.

---

## [Author Response]

Summary:In this manuscript, Sundquist, Hill, and colleagues present the 3.2 Å cryo-EM structure of yeast Vps4 bound to an ESCRT-III peptide. It builds on their paper published in eLife earlier this year that provided a 4.3 Å structure of Vps4 bound to nucleotides and a substrate peptide, and proposed a spiral staircase model for the continuous conveyance of substrate through the central pore. In this previous structure, the peptide density showed contacts with multiple subunits via their pore loops, but did not reveal side-chain density or the orientation of the bound peptide.A Krios data set now improved the resolution to 3.2 Å, which allowed the visualization of peptide side chains as well as pore loop contacts, and the proposal of a more detailed translocation model where every Vps4 subunit interacts with two amino acids through two "binding pockets" formed by the pore-1 and pore-2 loops. The new structure is thus consistent with the mechanisms the authors proposed previously, but does not reveal major new conceptual insights.

We have paid attention to wording in the revised manuscript in order to more clearly emphasize that our new 3.2Å structure is foundational and provides multiple major new conceptual insights, including:

1) The mechanism of substrate recognition is now revealed at the level of side chain interactions for Vps4 and presumably for the large family of related AAA+ ATPases that translocate proteins, which includes many important family members (e.g., proteasome). Our new structure therefore answers the long-standing mystery of how these proteins bind and process substrates in a sequence-independent manner.

2) Surprisingly, our new manuscript shows that Vps4 binds the tight-binding ESCRT-III-derived peptide in just one orientation. Moreover, the structure suggests a mechanism that would make this a general property for all substrates and possibly for some other related AAA+ ATPases. Rigorous testing of the generality of this conclusion will require substantial biochemical studies that are beyond the scope of the current manuscript, but the structural observation itself provokes a new way of thinking about AAA+ ATPases and will guide new biochemical studies.

3) Our new manuscript now reveals the identities of nucleotide – ATP (ADP·BeF_x_) or ADP – bound at each of the subunit active sites. This is fundamental to understanding the reaction cycle, which we now propose is driven by hydrolysis at the D subunit to destabilize the DE interface. As indicated by reviewer comments, this is a new concept for the AAA ATPase field that should also spur new lines of inquiry.

4) Many of the previously reported AAA+ ATPase structures have shown a six-fold rotationally symmetric arrangement of subunits, which is inconsistent with the implications of our structure. Moreover, none of the previously reported AAA+ ATPases structures have revealed substrate interactions at the level of the amino acid residue interactions that allows for a detailed mechanistic interpretation. Our new structure therefore challenges assumptions about multiple related proteins and adds reliability and detail to an alternative model.

5) A leading model for AAA+ ATPases that translocate proteins has been based upon studies of ClpX, which have been interpreted to indicate that the mechanism is non-sequential with subunits firing in random order. Our structure argues strongly against this mechanism, at least for Vps4 working under optimal conditions and probably for many other related enzymes too.

6) As indicated by reviewer comments, the dominant view of AAA mechanism has been that binding of different nucleotides (ATP vs. ADP) stabilizes different conformations of the subunits. The details present in our new structure prompt a different perspective in which subunit conformation is almost unchanged while changes in nucleotide change the stability of subunit interfaces.

The reviewers were a bit surprised to see that there are really not many new data in this submission albeit now having a structure at 1.1 Å higher resolution, and that the authors did not take advantage of this higher resolution to address and answer important outstanding questions.

The important point is not the amount of data but rather the amount of insight. As indicated above, we believe that our paper provides multiple conceptual advances that are foundational for understanding of mechanism and the design of future experiments.

Even though it is exciting to directly visualize substrate interactions of Vps4, the advance by this manuscript in its current form is rather limited and considered not sufficient for publication in eLife. The higher resolution data should enable the authors to pursue further functional exploration and reveal new conceptual insights that go beyond of what had already been established previously for Vps4 and other AAA motors. The manuscript thus needs major revisions before publication.

Our manuscript does provoke numerous new lines of enquiry, but exploring them rigorously would be beyond a scope that is reasonable for a single manuscript.

Essential revisions:1) The proposal of a directionality for ESCRT substrate translocation by Vps4 is interesting, but should be further supported by experimental data. Several related AAA motors have been shown to not have a preference for substrate orientation in the central pore, and for the conclusions here it should be considered that the described trapped structure likely represents an energy minimum, where the Vps4 subunits might arrange around the stably bound peptide. It is possible that a peptide bound in the opposite orientation could induce equivalent contacts, and that there is in fact no preference in the direction of translocation. The authors may be able to explore this through biochemical assays or structurally, for instance by using peptides of different lengths or with the inverted amino-acid sequence relative to the current one.

The fact that the peptide included in our structure binds Vps4 in one dominant orientation is a secure observation from our work. We agree that, like all experimentally determined structures, our structure represents an energy minimum, and that Vps4 has assembled around this peptide. However, the conditions used to prepare the Vps4-peptide complex would not discriminate between one or other of the two potential peptide orientations. As summarized in Figure 1—figure supplement 2, 109K particles fit structural classes that displayed good Vps4 features, and 82K of those particles were used in the final reconstruction, which clearly showed the peptide in just one orientation. If peptide binding in the opposite orientation was energetically similar to that observed, it would have been apparent in one of the classifications in our structural analysis.

We agree that the surprising observation of a unique (or at least strongly preferred) binding orientation for the peptide in our reconstruction raises important questions that merit further study. But they are beyond the scope of the current Research Advance manuscript.

2) The presented translocation model also does not go significantly beyond what had been previously discussed.

Our paper provides new insights on nucleotide state, changing subunit interactions, substrate side chain binding, and substrate orientation. Moreover, it is radically at odds with an alternative model of Vps4 mechanism that was published subsequently to our earlier paper.

The authors describe a correlation between nucleotide state and subunit conformation relative to the substrate, but the structural relationship remains unexplored in the manuscript. The impressive resolution of this structure should enable the authors to explain how ATP hydrolysis and release translates to specific rearrangements of the subunits or pore loops.

Our finding that nucleotide hydrolysis does *not* drive major conformational changes in the subunits or in the pore loops is an important conceptual advance that is only possible because of the relatively high resolution of our structure.

Reviewers were also not fully convinced by the authors' assignment of ADP- vs. ATP- (or ADP·BeF_x_) bound subunits, especially for the DE interface, and the presented model for ATPase-driven substrate translocation. As the authors know, reliably distinguishing between nucleotide states has been very difficult for other AAA hexamers in recent high-resolution cryo-EM studies, in part due to weaker density for beta and gamma phosphates, but also due to issues when averaging thousands of particles with potentially distinct nucleotide occupancies in some of the ATPase subunits. For the present study, it is indeed quite unexpected that neither the Mg ion nor the P-loop residues show significant changes in their orientation, and that the Arg fingers are just slightly shifted when comparing supposedly ADP·BeFx -bound and ADP-bound subunits. When discussing this surprising similarity of subunit interfaces, the authors bring up the possibility that their nucleotide assignments are wrong. The authors also try to come up with a rationale for how the very small structural differences may lead to preferential binding of ADP vs. ATP (or ADP·BeFx), but ignore in their Discussion that during normal motor function the nucleotide dictates subunit conformation, not the other way around.

We agree that it can be tricky to definitively distinguish between ATP (ADP·BeF_x_) and ADP even in a high quality 3.2Å cryo-EM map. That is why our manuscript notes the caveat that our assignment could be incorrect. Nevertheless, the density does seem clear in this regard and therefore compels the assignment given in the current model.

We understand that the reconstruction results from averaging of thousands of particles, each of which may be slight different, including some variation in bound nucleotide. Nevertheless, we do not see this as a fundamental concern. Even if, for example, a fraction of the particles have ADP·BeF_x_ bound at the D site, our mechanistic interpretation would still have to account for the binding of some ADP at that site as well as the even clearer interpretation that ADP is bound at the next (E) site in the reaction cycle.

We also agree that it is unexpected that upon binding of ADP the P-loop residues do not show significant changes in their orientation, and that the Arg fingers are just slightly shifted when comparing apparently ADP·BeF_x_-bound and ADP-bound subunits. As noted above, this finding underlies the important conceptual advance that the reaction cycle proceeds primarily because ATP hydrolysis changes the stability of a subunit interface rather than the conformation of a subunit.

Several biochemical and structural studies previously suggested major conformational and functional differences between ATP- and ADP-subunits, and that the transition from ATP- to ADP-bound through hydrolysis and phosphate release may drive the power stroke of AAA motors. Interestingly, the authors claim that substantial conformational adjustments are not required to pass through the DE states. But what then is the force-generating step for translocation, if subunits can transition from ATP to ADP-bound without any major conformational changes? ADP release or ATP binding to an empty Vps4 subunit? In a revised manuscript, the authors should assess and discuss which step of the ATPase cycle likely induces conformational changes and drives the conveyer belt for forceful substrate translocation. In that context it should also be discussed how constrains in the topologically closed ring could restrict conformational changes in response to hydrolysis, which may explain the authors' findings that ADP and ADP·BeF_x_-bound subunits don't show major conformational differences.

We have tried to clarify the presentation of our thinking in the revised manuscript. We agree that it is likely that the power stroke may result from ATP hydrolysis and phosphate release. Given the implication of our structure that hydrolysis occurs in the D subunit, the power stroke would be the release of phosphate that occurs upon transition to the E subunit conformation.

We do not completely understand the reviewer comments about a topologically closed ring, and so will address several possible points. (i) The six Vps4 subunits in our structure are each single polypeptide chains that are not covalently linked to each other. (ii) The Vps4 subunits are expressed as a fusion with the stable hexamer Hcp1, but as reported earlier our structure is of a construct that has a linker long enough to allow full ATPase and peptide-binding activities. (iii) As noted in the revised manuscript, the relatively conserved interactions of small domains around the periphery of the hexamer, and the reinforcing interactions of the Vta1 cofactor proteins, will inevitably influence the stability of interactions between large domains that define the nucleotide binding pockets, and hence the preference for binding of different nucleotides at those sites.

3) The reviewers have expressed concerns regarding the atomic model refinement. Only statistics for chains A-E are listed in Table 1, but the deposited models contain all 6 subunits in addition to the substrate. The validation report from the PDB for the full deposition reveals statistics that are much worse than those reported in Table 1, and the selective inclusion of only chain A-E is rather misleading.

This can be a tricky issue for structures such as ours in which some portions are well defined but others are at much lower local resolution. Specifically, the charge density map is generally well-defined for subunits A-E, but less so for subunit F or the Vta1 subunits. In light of this dramatic variation in local resolution, we believe that the correct approach is to use distinct approaches when building/refining different regions of the structure. Specifically, most of subunits A-E, have been refined by the all atom refinement approaches that are applicable for clearly defined regions of density. In contrast, subunit F and the Vta1 subunits have been built simply by docking of the previously determined subunit crystal structures and positioned by eye and rigid body refinement. In order to clarify this issue, we have added an explanatory note that the regions with poor statistics were simply docked in as previously determined crystal structures without further refinement. The statistics for subunits A-E reflect the quality of regions of the structure that are defined by the density and are germane to our mechanistic interpretation. The reason the whole structure displays apparently poor geometry is that the docking of crystal structures generates some steric clashes. In principle, these could be modeled away, but given the lack of clear information about how to rebuild those clashes and the distance of those regions from the features described in our manuscript, we have chosen not to manipulate the model further.